# Cellular Senescence and Periodontitis: Mechanisms and Therapeutics

**DOI:** 10.3390/biology11101419

**Published:** 2022-09-29

**Authors:** Sijia Chen, Dian Zhou, Ousheng Liu, Huan Chen, Yuehong Wang, Yueying Zhou

**Affiliations:** Hunan Key Laboratory of Oral Health Research, Hunan 3D Printing Engineering Research Center of Oral Care, Hunan Clinical Research Center of Oral Major Diseases and Oral Health, Xiangya Stomatological Hospital, Xiangya School of Stomatology, Central South University, Changsha 410013, China

**Keywords:** periodontitis, aging, cellular senescence, SASP

## Abstract

**Simple Summary:**

Oral health is increasingly recognized as an important part of overall health. Tooth loss is a contributor to overall musculoskeletal frailty and is closely associated with increased morbidity in the elderly population. Limited intervention exists to alleviate tooth loss associated with periodontitis, other than antibiotics. Mounting evidence suggests that targeting cellular senescence could slow down the fundamental aging process, and thus alleviate a wide range of age-related tissue dysfunctions, likely including tooth loss. Therefore, we feel it might be valuable to review the current understanding of the role and potential mechanisms of senescent cells in oral health with aging.

**Abstract:**

Periodontitis is a chronic inflammatory disease which increases in prevalence and severity in the older population. Aging is a leading risk factor for periodontitis, which exacerbates alveolar bone loss and results in tooth loss in the elderly. However, the mechanism by which aging affects periodontitis is not well understood. There is considerable evidence to suggest that targeting cellular senescence could slow down the fundamental aging process, and thus alleviate a series of age-related pathological conditions, likely including alveolar bone loss. Recently, it has been discovered that the senescent cells accumulate in the alveolar bone and promote a senescence-associated secretory phenotype (SASP). Senescent cells interacting with bacteria, together with secreted SASP components altering the local microenvironment and inducing paracrine effects in neighboring cells, exacerbate the chronic inflammation in periodontal tissue and lead to more alveolar bone loss. This review will probe into mechanisms underlying excessive alveolar bone loss in periodontitis with aging and discuss potential therapeutics for the treatment of alveolar bone loss targeting cellular senescence and the SASP. Inspecting the relationship between cellular senescence and periodontitis will lead to new avenues of research in this field and contribute to developing potential translatable clinical interventions to mitigate or even reverse the harmful effects of aging on oral health.

## 1. Introduction: Periodontitis and Aging

Periodontitis is caused by bacterial infection and progresses with inflammatory destruction of the tooth-supporting alveolar bone and soft tissue, which stems from complicated interactions between the subgingival microbiota and the host [1,2,3,4]. The 2017 World Symposium provided a more nuanced definition of periodontal disease staging and grading, and the discussion proposed four stages of periodontitis defined according to severity and complexity of management [4]. Clinical research shows that the prevalence and severity of periodontitis increase with aging [5,6,7,8]. It is quite common that moderate loss of alveolar bone and periodontal attachment occurs in the elderly [9]. In fact, aging causes most of the chronic diseases which restrict the quality of survival, independence, and prosperity [10]. Elderly populations exhibit increased susceptibility to several chronic disorders (cancers, atherosclerosis, diabetes, autoimmune or infectious diseases), including periodontitis [11,12]. Inextricably linked to aging, cellular senescence is the body’s reaction to DNA damage brought on by a variety of factors, including reactive oxygen species (ROS), telomere erosion, and other mitogenic and metabolic stresses [13]. However, it is highly controversial whether alveolar bone loss can be caused by aging alone in the absence of chronic periodontal inflammation [14]. The loss of alveolar bone and periodontal attachment as a result of aging is usually believed to be caused by a separate process; however, these changes are rather minor and of little clinical relevance. In other words, severe periodontitis is not a natural consequence of aging [15]. Yet it is worth noting that aging as a risk factor for periodontal disease affects periodontal tissues to exacerbate bone loss in elderly patients with periodontitis [9,12,14].

The underlying processes by which aging may impact periodontal inflammation and hence susceptibility to periodontitis have only been partially studied. Aging is associated with a state of low-grade “sterile” inflammation in the absence of overt infection [16]. Systemic chronic inflammation is a common hallmark of the aging process and has also been implicated in many age-related diseases, aggravating the pathology of these diseases. As a result, osteoporosis and other co-morbidities of aging are frequently observed to coexist with a periodontal disease of greater severity [17]. Other hypotheses suggest that the increased susceptibility to periodontitis in older individuals is the result of longer exposure to periodontal bacteria, shifts in subgingival microbiota composition, or age-related alterations in the immune response (immunosenescence) [12,14,18]. Immunosenescence is an age-related modification of both the adaptive and innate immunity, which impairs its appropriate function, also called “immune aging” [19,20]. It has been speculated that the absence of a well-regulated immune response contributes to older individuals developing an increased systemic chronic inflammatory state [16,21]. Recent studies show that both molecules and cells of the innate and adaptive immune response are adversely impacted by aging, combined with low-grade chronic inflammation, together to alter immunocompetence and promote the pathogenesis of a large number of diseases including in the oral cavity [22]. However, little is known about the source of the chronic inflammation that accelerates most major age-related diseases.

Cellular senescence is a fundamental mechanism of aging, which contributes to var-ious age-related conditions [23]. Cellular senescence refers to the essentially irreversible growth arrest, yet remains metabolically active and undergoes distinct phenotypic alterations [11,24,25,26]. Common markers of senescent cells include p16 and p21, while the discovery of new senescence markers suggests the increasing importance of senescence heterogeneity [27]. Senescent cells may be one of sources of chronic systemic inflammation: The senescence-associated secretory phenotype (SASP) is a group of proinflammatory cytokines, chemokines, and proteases secreted by senescent cells [28,29,30], which collectively alter the local environment [31,32,33]. Arrested cell proliferation, resistance to apoptosis and a complex SASP are three hallmarks of senescent cells [29]. Senescent cells accumulate in multiple tissues with aging [34,35]. Through their SASP, which is widely acknowledged as the probable link between senescence and age-related tissue failure, these cells can have significant deleterious impacts on tissue function [11,36,37]. In recent decades, the involvement of senescent cells in the pathophysiology of many age-related disorders has been widely researched, including life-shortening [38,39], cognitive decline [40], physical function decline [39,41,42], metabolic dysfunction [43,44,45], renal dysfunction [38], vasomotor dysfunction [46], atherosclerosis [47], stem cell dysfunction [48,49], joint degeneration [50,51,52] and osteoporosis [53]. However, the role of cellular senescence in disease onset, progression and remission in periodontitis is not well understood. Inspecting the link between cellular senescence and alveolar bone loss will provide new avenues of research in this field and help us to find potential paths to develop clinical interventions to reverse or mitigate the effects of aging on periodontitis.

## 2. Age-Associated Alterations in the Periodontal Microenvironment

Aging is linked to changes in tissues and cells as a result of the buildup of different chemicals and cellular damage over time, resulting in compromised homeostasis and a decreased ability to respond correctly to injuries or stresses. The anatomical and functional changes in periodontal tissues associated with the aging process have been well reviewed by Huttner [9]. It includes age-related alterations of osteoblasts and osteoclasts, dysregulated responses of periodontal tissue cells to the oral microbiota, and other common age-related biologic changes that can alter bone and tissue homeostasis. For example, lipopolysaccharide-stimulated PGE2, IL-1β, IL-6 and plasminogen activator production was lower in young human gingival fibroblasts than older cells [54]. The aged periodontal ligament cells produce increased quantities of PGE2 in response to forces caused by occlusal trauma, which may affect the severity of inflammation and subsequent tissue degradation [55]. In addition, histological studies have shown that compared with younger rats, the collagen density in the gingival tissues of old rats reduced and the degradation of collagen increased [56]. Another study confirmed this finding, revealing that aging causes gradual atrophy of the tooth-supporting tissues in rats [57]. These findings suggest that destruction of the periodontium in older individuals may be attributed to the progression of periodontitis as well as increased inflammatory changes and reduced tissue resilience induced by intrinsic alterations in aged periodontal tissues (Table 1).

Another explanation for the age-related degeneration of periodontal tissues could be cellular senescence [58]. The activation of the tumor-suppressor pathways, p16/retinoblastoma protein (Rb) and/or p53/p21, is linked to the establishment of “stable” growth arrest [11]. Therefore, cellular senescence is initially considered to be a mechanism of tumor suppression. However, mounting evidence indicates that accumulation of senescent cells in various tissues with advanced age could also negatively affect tissue function and homeostasis, and eventually lead to tissue pathology.

Age-related chronological senescence increases the number of senescent cells in many tissues [34,35]. Though they are relatively low in number (~1–10%), senescent cells cause tissue dysfunction through three major pathways: (i) The cellular function is dramatically impaired when cells become senescent. Thus, accumulation of senescent cells may impair tissue function in an autocrine manner [59]. (ii) Senescent cells could impair the function of neighboring cells either by making them senescent [39] or by damaging their differentiation potential [43,60]. Thus, senescent cells can impair tissue function in a paracrine manner. (iii) Regardless of how cellular senescence is induced, senescent cells are active in secreting SASP components, which potentially is one major mechanism for the etiology of age-related diseases and frailty [11]. The SASP constitutes a critical feature of senescent cells and mediates many of their pathophysiological effects. Happily, SASP factors mediate embryogenesis [61,62], wound healing [63], and activate immune responses to eliminate senescent cells [64,65]. Unfortunately, the SASP contributes to inflammaging (as cytokines and chemokines are significant features of the SASP) [22]. Furthermore, by distributing toxic substances to nearby bystander cells, the SASP can contribute to increased senescent cell accumulation and tissue dysfunction [30,31,39]. Thus, the SASP can explain some of the deleterious, pro-aging effects of senescent cells. In a word, cellular senescence has become a potential unifying mechanism of aging, age-related diseases, and frailty.

One plausible mechanism affecting periodontal tissues is the growth of senescent cells in the periodontal milieu with aging, specifically those cells that produce the SASP, which contributes to alveolar bone loss and enhancing susceptibility to periodontitis in the elderly. The load of senescent osteocytes rises in alveolar bone in a time-dependent way, according to a recent analysis [66]. Senescent osteocytes produce a pro-inflammatory substance called a SASP, which interacts with bacterial substances linked to periodontal inflammation and heightens the production of cytokines that are important in the development of inflammation. Additionally, these cytokines and matrix-degrading enzymes produced by senescent osteocytes transmit senescence to nearby cells in addition to harming the immediate microenvironment. Conditioned media from senescent osteocytes aggravate the lipopolysaccharide (LPS)-induced inhibition of osteocyte differentiation, decrease the migration of osteoprogenitor cells, and impair mineralization in vitro. These findings suggest that through influencing nearby osteoblast precursors, the SASP from senescent osteocytes may reduce bone regenerative potential in elderly persons [66]. Taken together, these findings suggest that the accumulation of senescent cells contributes to the deterioration of the periodontal environment by potentiating local inflammation induced by bacterial components, increasing extracellular matrix remodeling, and reducing regeneration in old age.

In the mouse skeleton, senescent osteocytes do not develop until around 18 months of age [67]. Unexpectedly, a significant number of defective senescent osteocytes (30%) were present at 6 months, probably as a result of prolonged exposure to periodontal pathogens and their toxins. These osteocytes appear to exist before the initiation of alveolar bone loss [68]. The bacterial-derived LPS not only plays a role in the obtaining of “premature” cellular senescence, but also may promote the secretion of the SASP that aggravates localized inflammation, eventually leading to alveolar bone loss [68]. It is important to know that LPS is not the only factor associated with periodontal disease; other virulence factors could also influence the production of proinflammatory cytokines that activate osteoclastogenesis. More bacterial products in dental disease need to be further explored. The abnormally high senescent cell burden in young alveolar bone may be a new pathogenic mechanism that contributes to oral bacteria promoting inflammatory alveolar bone destruction.

## 3. Cellular Senescence and Immune Function

Periodontitis is a chronic inflammatory disease in which tissue damage results from dysregulated and long-term inflammatory responses to the persisting subgingival biofilm. In order to determine if the aging-related change in immune function may be relevant to the etiology of periodontitis, consideration of this issue is important.

Recently, in an in vitro experiment, Porphyromonas gingivalis (Pg) was found to infect dendritic cells (DCs), activate related SASPs (e.g., IL-1β, IL-6, IL-8) and exosomes (EXO), and induce DCs senescence, while affecting surrounding DCs through secreted SASPs and EXO, expanding the senescence range and affecting periodontitis [69]. And the success of DCs exosomes in the treatment of alveolar bone degeneration suggests a correlation between alveolar tissue aging and immunity [70]. In vivo mechanistic studies in mice revealed that the senescence biomarkers beta galactosidase (SA-β-Gal), p16, p21, IL6, TNFα and IL1β expression were elevated in aged and periodontitis, and that bone marrow-like CD11c+ and T cells were prone to senescence in vivo, with Pg-induced EXO of DCs being the main causative agent of alveolar bone loss and immune senescence. These experimental results suggest that periodontitis is associated with and exacerbated by immune senescence with old age, and presumably its secretion of associated SASPs creates an environment conducive to inflammation and bacterial development in mice, promoting oral flora dysbiosis and causing accelerated alveolar bone degeneration [71]. The relationship between aging and immunity in periodontitis remains to be further investigated, and it is now speculated that periodontitis may be associated with immune aging. Since age-associated innate and adaptive immunity alterations in periodontal diseases have been extensively reviewed [12,14,72,73], in this section we will look at how cellular senescence and SASPs affect immune cell destiny and function.

The immune system is crucial in the fight against microbial infections. However, there is evidence that as people age, both the innate and adaptive arms of the immune system may change, making them more susceptible to infection [12,74]. Age-related changes in immune function might not be necessarily equivalent to immunodeficiency, but to dysregulation of the immune response [75,76]. Such changes most likely result in a decreased ability to regulate infections, which promotes chronic pathogen persistence and increases the number of defense cells in the periodontal tissues. This can then result in additional tissue damage and illness [76].

The expression and function of senescence markers such as p16 in murine bone marrow-derived macrophages (BMDM) and human adipose tissue macrophages are well understood, as is the scientific development [77,78]. In vitro, the levels of p16 expressed by IL-4-polarized human M2 macrophages were lower than IFN-γ-polarized M1 [77,78]. Furthermore, in murine and human macrophages, p16 expression suppresses LPS-induced IL-6 production [79]. Therefore, p16, as a cellular senescence marker, is involved in the differentiation of monocytes into inflammatory macrophages.

The expression levels of both p16 and p14/p19 increased with aging in all B lineages, especially in pro-B, pre-B, and IgM+ mature B cells [80,81]. Ectopic expression of p16 or p14/p19 in young pro-/pre-B cells mimics the effect of aging by reducing cell growth and survival. Downregulation of the CDKN2A gene, on the other hand, boosts these cells’ proliferative ability [82]. Individuals’ proportions of senescence-like CD4+ T cells tend to rise with age [83,84]. Senescent T cells are metabolically active, producing a variety of cytokines. Non-senescent cells can be affected negatively or positively by pro-inflammatory substances found in the SASP of senescent T cells. This inhibition is thought to be a pro-tumor mechanism since it necessitates intercellular interaction [85].

Cellular senescence is found in hematopoietic progenitors as well as in particular specialized immune cells. Studies have shown that murine hematopoietic stem cells (HSCs) accumulate DNA damages and senescence markers with aging [86,87]. As a result, immune system homeostasis is impacted by replicative and age-induced HSC senescence.

In addition to the inherent senescence of immune cells, immune cells are also modulated by their aging microenvironment. The SASP may interact with the immune system extensively. The SASP promotes the migration of immune cells that help remove senescent cells [88]. This secretome influences the types and amounts of immune cells recruited, as well as their cell fate determination. During liver fibrosis, for example, p53-expressing senescent liver satellite cells produce IFN- and IL-6, skewing the polarization of resident Kupffer macrophages and newly recruited macrophages toward the proinflammatory M1 phenotype [89]. However, tissue macrophage response appears to wane with age, which may contribute to the accumulation of senescent cells in old age. Immune surveillance may be hampered by senescent cells interfering with immune activity [90]. Chronic IL-6 exposure reduces macrophage activity, and SASP proteases may degrade the FAS ligand or other cell surface proteins necessary for efficient immunological function, supporting this idea [91]. When young macrophages were challenged with aged serum, macrophages reduced secretion of TNFα and increased basal levels of IL-6 [92]. Furthermore, injecting young peritoneal macrophages into the peritoneal cavities of elderly mice resulted in decreased phagocytosis and elevated T and B cell numbers [93]. This suggests that the aging microenvironment has a significant impact on macrophage function. More mechanistic insights are needed to understand how senescent cells and their SASP affect the immune system as well as whether lowering the senescent cell load improves immune responses to infections.

These data show that aging affects the immune system by impairing progenitor self-renewal and shifting pluripotency toward myeloid lineages. Additionally, an increase in the number of senescent cells in the immune system and tissues will result in the age-related decrease in tissue function. Immune cells can be controlled in a variety of ways, which may be part of a more dynamic interplay between intrinsic senescence processes, the aging milieu, and other cell types in the surrounding area.

## 4. Cellular Senescence and Bone Metabolism

One cause of bone loss during aging is impaired homeostasis of bone metabolism. Alveolar bone is highly active in metabolism and remodeling, and is the main structure supporting teeth in periodontal tissue. In the pathophysiology of bone loss with age, complicated interactions between cells of the osteoclast and osteoblast lineages play a significant role [94]. The former is a supply of cells for bone repair, whereas the latter represents inflammatory processes and bone resorption. Both lineages are weakened with aging due to a combination of inherent and environmental influences, resulting in a considerable loss in osteogenic capacities [95].

The non-human primate model shows that the gingival transcriptome environment in aging healthy tissues reflects a more general osteogenic homeostasis and the potential for regulating osteoclast and osteoblast functions to control dysregulated microbial stimulation from the accretion of oral biofilms [96]. Changes in gene expression in periodontitis, especially in aged animals, were skewed toward creating an environment with substantial osteoclastogenic potential consistent with increased bone resorption in periodontitis [72]. These data indicate that local bone resorption is up-regulated under the condition of aging, creating a destructive environment.

This new work extracts and analyzes ex vivo highly enriched populations of bone and hematopoietic lineage cells from mouse bone marrow. It shows that a range of cell types in the bone microenvironment become senescent with chronological age, and that senescent myeloid cells and senescent osteocytes are the major sources of the SASP [67]. Either genetic or pharmacological means to eliminate senescent cells in the bone microenvironment could prevent age-related bone loss in mice. This study additionally demonstrates that the senescent-cell conditioned medium (containing SASP) impaired osteoblast mineralization and enhanced osteoclast-progenitor survival in vitro, leading to increased osteoclastogenesis [53]. These data collectively establish a causal role for senescent cells and their SASP in skeletal bone loss with aging, and demonstrate that targeting senescent cells has both anti-resorptive and pro-anabolic effects on bone.

## 5. Cellular Senescence and Healing Capacity

Various studies have identified obvious defects in the wound healing of aged periodontal tissues [97,98,99]. Changes in the susceptibility to periodontitis with aging could be explained by exposure to pro-inflammatory conditions and changes in the healing capacity of cells and tissues [9]. In response to signals from injured tissues, mesenchymal stem cells (MSCs) can migrate from the perivascular region into the blood circulation. MSCs circulating in the bloodstream may then collect in damaged tissues and assist in the regeneration process [100,101]. However, aged MSCs contain more senescent cells [41], which have a negative impact on their immunomodulatory and differentiation capacities [102]. Stolzing et al. [103] found that the number and cell proliferation of MSCs extracted from older humans decreased, and their capacity to undergo osteogenic differentiation was impacted. Aged mice presented fewer PDGFRα+ MSCs compared to young mice after ligation, with an increase in the amount of inflammatory T and B cells at the periodontitis site [104]. The decreased immune tolerance and increased bone degradation at the periodontitis site may be linked to the functional impairment of older MSCs.

Osteoblast precursor recruitment is necessary for bone reconstruction following bone resorption, and senescent cells have a potent paracrine influence on the cells around them. It has been demonstrated that senescent osteocytes can produce substances that inhibit the recruitment of osteoprogenitor cells, hence interfering with the process of bone production [66]. In addition, continuous exposure to SASP factors and continual cell replacement to repair injured osteocytes throughout life may possibly propagate senescence toward progenitor cells, hastening their premature exhaustion, reducing their proliferative capacity, and causing problems in alveolar bone regeneration [68]. Thus, aging-induced phenotypic alterations in MSCs or osteoprogenitor cells may be linked to a decline in periodontal tissue regeneration capabilities in older people (Figure 1).

## 6. Interventions Targeting Senescent Cells and the SASP

In light of osteocytes orchestrating bone remodeling, the high number of dysfunctional senescent osteocytes and other types of senescent cells in alveolar bone may compromise tissue homeostasis. Therefore, targeting senescent cells could be a novel approach to alleviate alveolar bone loss.

Interfering with pathways that lead to senescence, eliminating senescent cells, and targeting the SASP to minimize the detrimental consequences of senescent cells are all being studied as techniques to attenuate the adverse effects of senescent cells [11]. The first strategy is to inhibit the formation of senescent cells through targeting the cell stresses or signaling pathways which lead to senescence-associated growth arrest. However, interfering with tumor suppressor pathways such as Rb, p16^INK4A^ or p53, could compromise fundamental anti-cancer mechanisms and be likely to promote cancer [105,106].

The second method, which involves removing senescent cells that have already developed, could help to minimize tissue inflammation and organ malfunction as people age. Senescent cells’ unique shape, secreted protein patterns, and gene expression profiles indicate the feasibility of this method [107]. Genetic clearance of p16^Ink4a^ (i.e., the INK-ATTAC ”suicide” transgene encoding an inducible caspase 8 expressed specifically in senescent cells [108]) has been demonstrated to help with age-related health and longevity [108], as well as osteoporosis [53]. In physiological aging, the pharmacological clearance of senescent cells with a senolytic medication cocktail containing dasatinib and quercetin is also beneficial in repairing bone integrity [53]. Clinical trials evaluating the effectiveness of senolytic medicines in the treatment of age-related comorbidities, such as osteoporosis, are either underway or planned. The capacity to selectively target senescent cells with biological or small chemical “senolytic” therapy without generating serious side effects is necessary for the translation of these results to patients.

The third strategy aims to stop the SASP from developing or lessen its consequences. The possibility exists to disrupt the SASP without affecting the anti-oncogenic pathways engaged in senescent cells [11,109]. For example, inhibiting the JAK-STAT pathway suppresses the SASP in preadipocytes and endothelial cells as well as SASP-induced adipose tissue inflammation in vitro, and also attenuates age-related adipose tissue and systemic inflammation together with frailty in vivo [30]. JAK inhibitors and ruxolitinib can help with age-related osteoporosis by suppressing particular factors including IL-6, IL-8, and PAI-1, which have been proven to promote osteoclast production [53]. A range of pharmacological therapies, including metformin, rapamycin, and NFB inhibitors, have been demonstrated to reduce the SASP in addition to JAK inhibitors [26].

Targeting senescent cells is likely to be beneficial in the treatment of alveolar bone loss in periodontitis with age, making it fundamentally distinct from all other periodontal treatments now available. Eliminating senescent cells and/or inhibiting their proinflammatory secretome also improves the cardiovascular function [46], alleviates osteoporosis [53] and reduces frailty [30] in old mice. Thus, targeting cellular senescence represents a novel therapeutic strategy to prevent not only alveolar bone loss but potentially multiple age-related diseases simultaneously.

## 7. Challenges Ahead

To clarify the biological mechanisms by which aging can affect periodontitis, several key difficulties need to be overcome.

Firstly, it is difficult to distinguish the effects that are caused by environmental factors or intrinsic aging, due to the direct contact of the oral cavity with the external environment and risk factors [110]. Secondly, studies in mice are not always conserved for humans. Many important discoveries in mice have been successfully translated to humans, but many others have not. This may be due to inherent biological differences between mice and humans [111]. Besides, in vitro studies may not always be relevant to the in vivo setting. The aging-associated in vivo microenvironment may further contribute to the complexity of both innate immune and adaptive functional deficits [12]. In vivo, these innate and adaptive immune cells interact with stromal cells in the periodontal tissues, leading to different outcomes under similar stimulation [76]. Thirdly, the genetic heterogeneity among human subjects is a challenge for any type of biomedical research. Humans are characterized by great genetic heterogeneity, which plays a key role in disease susceptibility, lifespan and an individual’s response to drugs [111].

Lastly, the interaction of aging and inflammation is a highly complex process without well-understood causality or directionality. Currently, it would be difficult to investigate whether age-related processes have caused or resulted from inflammation, or whether they are bidirectional [14].

## 8. Summary

In conclusion, the prevalence and severity of periodontitis increase with aging. The elucidation of cellular senescence in the periodontal microenvironment may help to understand the role of aging in periodontitis. Senescent cells such as osteocytes are abundant in the aged periodontium, and can damage local microenvironment and influence neighboring cells through the SASP. In addition, senescent cells could interact with subgingival bacteria to exacerbate persistent inflammation in old age. As a result of a combination of intrinsic senescence and an aging milieu, cells and chemicals in the periodontium that are involved in immune response, bone metabolism, and tissue regeneration were disrupted, which eventually leads to the further development of periodontitis. Targeting cellular senescence to prevent excessive alveolar bone loss holds promise for a novel treatment strategy for periodontitis.

## Figures and Tables

**Figure 1 biology-11-01419-f001:**
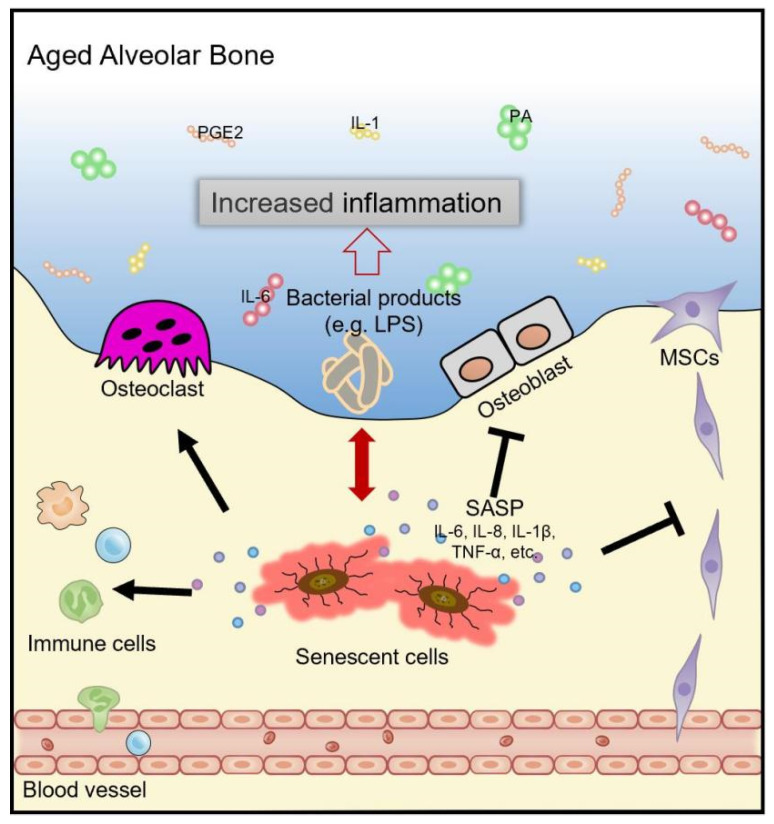
Mechanisms underlying excessive alveolar bone loss in periodontitis with aging. Due to advancing age or long-term exposure to bacterial products (e.g., LPS), the senescent osteocytes accumulate in the alveolar bone and develop a proinflammatory SASP (IL-6, IL-8, IL-1β, TNF-α, etc.), which can exacerbate the chronic inflammation of periodontal tissue. Pro-inflammatory cytokines and matrix-degrading enzymes secreted by senescent cells can damage the local microenvironment and induce bystander effects in neighboring cells. Owing to intrinsic senescence and the aging microenvironment, the dysregulation of immune responses may lead to the chronic persistence of pathogens and increased accumulation of immune cells in the periodontal tissues. Consequently, the homeostasis between osteoblasts and osteoclasts is disrupted, and the immunomodulatory effects, migratory ability and differentiation potential of MSCs are impaired, both of which lead to defective alveolar bone regeneration. All of these factors could result in more alveolar bone loss in periodontitis with aging.

**Table 1 biology-11-01419-t001:** The effects of aging on periodontium.

	Age-Related Changes
Tissue	Gingiva: a thinning of the epithelium; diminished keratinization. Periodontal ligament: the fiber and cellular contents decrease; uneven and irregular Sharpey’s fibers insertions. Cementum: cementum volume increases. Alveolar bone: bone formation steadily declines and loss of bone mass.
Cell	Gingival fibroblasts (GFs): collagen production decreases; release more inflammatory cytokines such as prostaglandin E2 (PGE2), interleukin (IL)-1, IL-6 and plasminogen activator (PA); a reduction in mRNA levels; an increase in the production of PGE2, Cox-2 and IL-1 mRNA. Periodontal ligament cells: a significant reduction in chemotaxis, motility, proliferation rates and differentiation ability; a decrease in osteoblast proliferating precursors or synthesis and secretion of essential bone matrix proteins; an increase in the production of PA, PGE2, IL-1, and IL-6; enhanced RANK expression on osteoclast progenitors and RANKL expression in the mesenchymal stromal cells. Immune cells: immune senescence occurs and the secretion of associated SASP creates an environment conducive to inflammation and bacterial development, promoting dysbiosis of the oral flora and leading to accelerated degeneration of the alveolar bone.
Molecule	Type I collagen decreases; increased IL-1, PA, and the plasminogen activator inhibitor-2 (PAI-2) release in gingival crevicular fluid.

## Data Availability

Not applicable.

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
