# Peer review of "Cellular Senescence and Periodontitis: Mechanisms and Therapeutics"

_biology, 2022, doi:10.3390/biology11101419_

Round 1

Reviewer 1 Report

This is a very interesting and important review to the field of periodontics. The authors in this study propose mechanisms that may explain the increased progression of periodontitis in aging population. This is a hot topic given the high prevalence of periodontitis  in population above 65 age.The authors mentioned the immunosenescence  and the associated SASP  in aging population as a the contributing factor. However, they were more  focused on relating the effect of SASP  derived form senescent osteocytes. They ignored  recent important studies showing the role of immunosenscence and SASP from key immune cells like dendritic cells. SASP includes free pro-inflammatory cytokines/ factors and other kind of secretomes  such as extracellular vesicles and Exosomes. The role of extra-cellular  vesicles and Exosomes  is  not discussed in-depth. These need to be discussed and added for comprehensive explanation of the senescence related mechanism that links periodontitis and aging .Please discuss and mention the findings of the following studies highlighting the role of SASP including pro-inflammatory Exosomes in periodontitis.

-Ranya Elsayed Mahmoud Elashiry Yutao Liu Ana C. Morandini Ahmed El-Awady Mohamed M. Elashiry Mark Hamrick Christopher W. CutlerMicrobially-Induced Exosomes from Dendritic Cells Promote Paracrine Immune Senescence: Novel Mechanism of Bone Degenerative Disease in Mice. Aging and disease. 2022 https://doi.org/10.14336/AD.2022.0623

-Elashiry M, Elashiry MM, Elsayed R, Rajendran M, Auersvald C, Zeitoun R, Rashid MH, Ara R, Meghil MM, Liu Y, Arbab AS, Arce RM, Hamrick M, Elsalanty M, Brendan M, Pacholczyk R, Cutler CW. Dendritic cell derived exosomes loaded with immunoregulatory cargo reprogram local immune responses and inhibit degenerative bone disease in vivo. J Extracell Vesicles. 2020 Aug 7;9(1):1795362

Author Response

We deeply appreciate reviewer 1’s positive comments on our manuscript. We completely agree that the contribution of other senescent cells and exosomes is very valuable to discuss in the current review. We now have cited a number of important studies (including two papers mentioned above) and discussed them in line 184 -199.

Reviewer 2 Report

It is important to correct that periodontitis is not a chronic infectious disease, is a chronic inflammatory disease, so I suggest you update this information according to what is reported in the new classification. (Tonetti MS, Greenwell H, Kornman KS. Staging and grading of periodontitis: Framework and proposal of a new classification and case definition [published correction appears in J Periodontol. 2018 Dec; 89 (12):1475]. J Periodontol. 2018;89 Suppl 1:S159-S172. https://doi.org/10.1002/JPER.18-0006 PMID: 29926952)

The review is interesting, however despite the fact that several references are included that indicate the importance of SASP as indicated in the following sentence “Senescent cells might be one of sources for the chronic systemic inflammation: The senescence-associated secretory phenotype (SASP) is a group of proinflammatory cytokines, chemokines, and proteases secreted by senescent cells[26-28], which 79 collectively alter the local environment[29-31]”

It would be very good to indicate what these cytokines and chemokines are and include them in Figure 1, also that not only LPS is the virulence factor associated with the production of proinflammatory cytokines that activate osteoclastogenesis.

I think it is important to include these virulence factors associated with the development of periodontitis

Reviewer 3 Report

Dear Authors,

1. The research by Chen S et al focuses on the role of senescence in oral health and especially on periodontitis.  2. The topic of the paper is very interesting taking into consideration that periodontitis incidence increases with age and while there is significant research on the topic, little clinical progress has been achieved. Therefore, it is important to identify all the factors involved and find potential new treatment targets.  3. The article presents the updated knowledge on the subject.  4. Given that the manuscript is a narrative review, in my opinion (as I suggested to the authors in my comments), a table would be needed to summarise the effects of senescence in periodontal tissues. 5. The conclusions are consistent with the evidence and arguments presented and the authors address the main question posed. 6. The references are appropriate. 7. The figure is conclusive. A table would be helpful.

Congratulations for your work!

Best regards

Author Response

We deeply appreciate the positive comments from this reviewer. We believe adding a table would be an excellent idea. We now have added a new table (Table 1) to summarize the effects of aging/senescence on periodontal tissues on 3 levels (tissue, cell and molecule). 

Reviewer 4 Report

Tooth loss is a contributor to overall musculoskeletal frailty and is tightly associated with increased morbidity in an elderly population. Limited intervention exists to alleviate tooth loss associated with periodontitis, other than antibiotics. Mounting evidence suggests that targeting cellular senescence could slow down the fundamental aging process and thus alleviate a wide range of age-related tissue dysfunction, likely including tooth loss. The review systematically reviews the concept of senescence and periodontics progress.

Strong points: It is good to point out that targeting tumor suppressor pathways, such as Rb, p16INK4A, or p53, would compromise fundamental anticancer mechanisms and is likely to promote cancer.

The authors provide insightful opinions in part 7. They are great thoughts and worthy of being raised up.

Weak points:  Define the concept of Cellular senescence at the beginning. It should not be in line 114.

For the most part 3, Cellular senescence and Immune function are not periodontics related. It is more like a general review of senescence and the immune system. It would be better with more periodontics literature.   

Author Response

We thank this reviewer for pointing out the strong points for our review. We also agree that it will be better to define cellular senescence at the beginning, and thus have moved it to line 50-52. We also cited several additional references to discuss the immune regulation in periodontal diseases in line 184-199. To note, there are very few studies published to examine the role of senescent cells in periodontal diseases, especially for in vivo studies. This is one of reasons why we feel our review might be valuable and timely for the field.